# Gradient Dynamics of Shallow Univariate ReLU Networks

**Francis Williams**[*]            **Matthew Trager**[*]

**Claudio Silva**        **Daniele Panozzo**        **Denis Zorin**        **Joan Bruna**

New York University

## Abstract

We present a theoretical and empirical study of the gradient dynamics of overparameterized shallow ReLU networks with one-dimensional input, solving least-squares interpolation. We show that the gradient dynamics of such networks are determined by the gradient flow in a non-redundant parameterization of the network function. We examine the principal qualitative features of this gradient flow. In particular, we determine conditions for two learning regimes: *kernel* and *adaptive*, which depend both on the relative magnitude of initialization of weights in different layers and the asymptotic behavior of initialization coefficients in the limit of large network widths. We show that learning in the kernel regime yields smooth interpolants, minimizing curvature, and reduces to *cubic splines* for uniform initializations. Learning in the adaptive regime favors instead *linear splines*, where knots cluster adaptively at the sample points.

## 1   Introduction

An important open problem in the theoretical study of neural networks is to describe the dynamical behavior of the parameters during training and, in particular, the influence of the dynamics on the generalization error. To make progress on these issues, a number of studies have focused on a tractable class of architectures, namely single hidden-layer neural networks. For a fixed number of neurons, negative results establish that, even with random initialization, gradient descent may be trapped in arbitrarily bad local minima [27, 31], which motivates an asymptotic analysis that studies the optimization and generalization properties of these models as the number of neurons $m$ grows.

Recently, several works [13, 2, 6, 12, 23] explained the success of gradient descent at optimizing the loss in the *over-parameterized* regime, *i.e.*, when the number of neurons in significantly higher than the number of training samples. In parallel, another line of work established global convergence of gradient descent using tools from optimal transport and mean-field theory [8, 26, 22, 29]. The essential difference between these two approaches was pointed out in [7], and is related to the use of a different scaling parameter as the number of neurons tends to infinity: in one case, the neural network behaves asymptotically as a kernel machine [19], which in turn implies that as over-parameterization increases, the parameters stay close to their initial value; in contrast, in the mean field setting, parameters asymptotitically evolve following a PDE based on a continuity equation.

Although both scaling regimes explain the success of gradient descent optimization on overparametrized networks, they paint a different picture when it comes to generalization. The generalization properties in the kernel regime borrow from the well established theory of kernel regression in Reproducing Kernel Hilbert Spaces (RKHS), which has been applied to kernels arising from neural networks in [17, 14, 20, 24, 10], and provide a somehow underwhelming answer to the benefit of neural networks compared to kernel methods. However, in practice, large neural networks do not exhibit

---

[*]Equal contribution.

the traits of kernel/lazy learnings, since filter weights significantly deviate from their initialization despite the over-parameterization. Also, empirically, active learning provides better generalization than kernel learning [7], although the theoretical reasons for this are still poorly understood.

In this paper, we consider a simplified setting, and study wide, single-hidden layer ReLU networks defined with one-dimensional inputs. We show how the kernel and active dynamics define fundamentally different function estimation models. For a fixed number of neurons, the network may follow either of these dynamics, depending on the initialization. Specifically, we show that kernel dynamics correspond to interpolation with *cubic splines*, whereas active dynamics yields *adaptive linear* splines, where neurons accumulate at the discontinuities and yield piecewise linear approximations.

**Further related work.**   Our work lies at the intersection between two lines of research: the works, described above, that study the optimization and generalization for shallow neural networks, and the works that attempt to shed light on these properties on low-dimensional inputs. In the latter category, we mention [4] for their study of the abilities of ReLU networks to approximate low-dimensional manifolds, and [32] for their empirical study of 3D surface reconstruction using precisely the intrinsic bias of SGD in overparametrised ReLU networks. Another remarkable recent work is [11], where the approximation power of deep ReLU networks is studied in the context of univariate functions. Our analysis in the active regime (Sec. 3.1) is closely related to [21], in which the authors establish convergence of gradient descent to piece-wise linear functions under initializations sufficiently close to zero. We provide an Eulerian perspective using Wasserstein gradient flows that simplifies the analysis, and is consistent with their conclusions. The implicit bias of SGD dynamics appears in several works, such as [30, 15], and, closest to our setup, in [28], where the authors observe a link between gradient dynamics and *linear splines*. They do not however observe the connection with *cubic splines*, although they observe experimentally that the function returned by a network is often smooth and not piecewise linear. Finally, we mention related work that studies the tesselation of ReLU networks on the input space [16].

**Main contributions.**   The goal of this paper is to describe the qualitative behavior of the dynamics or 1D shallow ReLU networks. Our main contributions can be summarized as follows.

• We investigate the gradient dynamics of shallow 1D ReLU networks using a "canonical" parameterization (Sec. 3.1). We show that the dynamics in this case are are completely determined by the evolution of the residuals. Furthermore, neurons will always accumulate at certain sample points where the residual is large and of opposite sign compared to neighboring samples. This means that the dynamics in the reduced parameterization biases towards functions that are *piecewise-linear*.

• We observe that the dynamics in full parameters are related to the dynamics in canonical parameters by a change of metric that depends only on the network at initialization. This change of metric is expressible in terms of invariants $\delta_i$ associated with each neuron. When $\delta_i \gg 0$ the dynamics in full parameters (locally) agree with the dynamics in reduced parameters; when $\delta_i \ll 0$, the dynamics in full parameters (locally) follow *kernel dynamics*, in which only the outer layer weights change.

• We consider the idealized kernel dynamics in the limit of infinite neurons, and we show that the RKHS norm of a function $f$ corresponds to a weighted L2 norm of the second derivative $f''$, *i.e.*, a form of *linearized curvature*. For appropriate initial distributions of neurons, the solution to kernel learning is a smooth *cubic spline* (Theorem 5). This illustrates the qualitative difference between the "reduced" and "kernel" regimes, which depend on parameter lift at initialization. Arbitrary initializations will locally interpolate between these two regimes.

• We also discuss the effect of applying a scaling parameter $\alpha(m)$ the network function (where $m$ is the number of neurons), which becomes important as the number of neurons tends to infinity. As argued in [7], when $\alpha(m) = o(m)$, the variation of each neuron will asymptotically go to zero (*lazy regime*), so our local analysis translates into a global one.

## 2   Preliminaries

We consider the problem of training of a two-layer ReLU neural network with $m$ scalar inputs and a single scalar output using the least-squares loss:

$$\min_{\boldsymbol{z}} L(\boldsymbol{z}) = \frac{1}{2} \sum_{j=1}^{s} |f_{\boldsymbol{z}}(x_j) - y_j|^2$$

$$\text{where} \quad f_{\boldsymbol{z}}(x) = \frac{1}{\alpha(m)} \sum_{i=1}^{m} c_i [a_i x - b_i]_+, \quad \boldsymbol{z} = (\boldsymbol{a} \in \mathbb{R}^m, \boldsymbol{b} \in \mathbb{R}^m, \boldsymbol{c} \in \mathbb{R}^m).$$

(1)

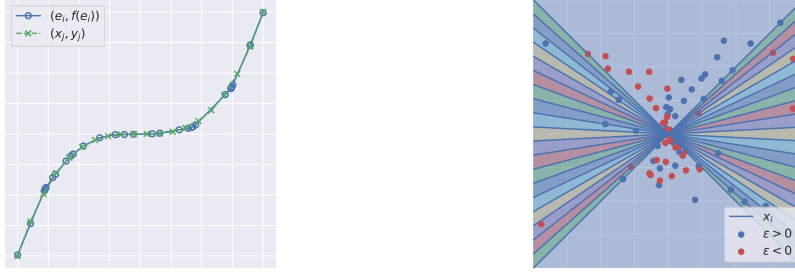

Figure 1: *Left:* A network function $f_{\boldsymbol{z}}(x)$ interpolating input samples (blue x's). The knots of $f_{\boldsymbol{z}}(x)$ as a piecewise linear function are plotted as green circles. *Right:* The canonical parameters of the network visualized as in (6). Each particle represents a neuron and the color indicates the sign of $\epsilon_i$. The samples $x_j$ correspond to lines $ux_j + v = 0$. The colored regions which correspond to different activation patterns of neurons on the training data.

Here $(x_i, y_i) \in \mathbb{R}^2$, $i = 1, \ldots, s$ is a given set of samples, $\boldsymbol{z}$ is a vector of parameters, and $\alpha(m)$ is a normalization factor that will be important as we consider the limit $m \to \infty$. We are interested in the minimization of (1) performed by gradient descent over the parameters $\boldsymbol{z}$. This scheme may be analyzed through its continuous-time counterpart, the gradient flow

$$\boldsymbol{z}(0) = \boldsymbol{z}_0, \qquad \boldsymbol{z}'(t) = -\nabla L(\boldsymbol{z}(t)). \tag{2}$$

While (2) describes the dynamics of $\boldsymbol{z}(t)$ in parameter space, we are ultimately interested in the trajectories of the function $f_{\boldsymbol{z}(t)}$. Let $\mathcal{F} := \{f : \mathbb{R} \to \mathbb{R}\}$ denote the space of square-integrable scalar functions, and let $\varphi$ be the function-valued mapping $\varphi(\boldsymbol{z}) := f_{\boldsymbol{z}}$. Since $L(\boldsymbol{z}) = R(\varphi(\boldsymbol{z}))$ with $R(f) = \frac{1}{2} \sum_{j \le s} |f(x_j) - y_j|^2$, we have by the chain rule that the dynamics of $g(t) := \varphi(\boldsymbol{z}(t)) = f_{\boldsymbol{z}(t)}$ are described by

$$g(0) = f_{\boldsymbol{z}_0}, \qquad g'(t) = -\nabla\varphi(\boldsymbol{z}(t))^\top \nabla\varphi(\boldsymbol{z}(t)) \nabla R(g(t)) . \tag{3}$$

The dynamics in function space are thus controlled by a time-varying *tangent kernel* $K_t := \nabla\varphi(\boldsymbol{z}(t))^\top \nabla\varphi(\boldsymbol{z}(t))$. It was shown in [19] that under certain assumptions the kernel $K_t$ remains nearly constant throughout training.

It is immediate to see that the parameters $\boldsymbol{z}$ can be continuously rescaled without affecting the function $f_{\boldsymbol{z}}$, according to $(a_i, b_i, c_i) \mapsto (a_i k_i, b_i k_i, c_i / k_i)$ with $k_i > 0$. In order to eliminate this ambiguity, we introduce the following *canonical parameterization* of the network's functional space:

$$\tilde{f}_{\boldsymbol{w}}(x) = \frac{1}{m} \sum_{i=1}^{m} r_i \langle \tilde{x}, d(\theta_i) \rangle_+, \qquad \boldsymbol{w} = (\boldsymbol{r} \in \mathbb{R}^m, \boldsymbol{\theta} \in [0, 2\pi]^m), \qquad \tilde{x} = (x, 1). \tag{4}$$

where $d(\theta_i) = (\cos\theta_i, \sin\theta_i) \in S^1$. The natural mapping into canonical parameters is given by

$$\pi(a_i, b_i, c_i) = \left( \frac{m}{\alpha(m)} c_i \sqrt{a_i^2 + b_i^2}, \arctan(-b_i/a_i) \right) = (r_i, \theta_i). \tag{5}$$

This mapping clearly satisfies $\tilde{f}_{\pi(\boldsymbol{z})} = f_{\boldsymbol{z}}$. We can also define the loss with respect to this parameterization as $\tilde{L}(\boldsymbol{w}) = L(\boldsymbol{z})$ where $\boldsymbol{w} = \boldsymbol{\pi}(\boldsymbol{z})$. We will compare the dynamics of $L(\boldsymbol{z})$ with those of $\tilde{L}(\boldsymbol{w})$ to study the impact on training of different choices of parameterization and initialization, as well as the asymptotic behavior of (3) as $m$ increases.

**Visualizing a network function.** We can visualize a network function $f_{\boldsymbol{z}}(x)$ in two ways. First, we can plot $f_{\boldsymbol{z}}(x)$ as a scalar function (Figure 1, left). Note that $f_{\boldsymbol{z}}(x)$ is a continuous piecewise linear functions in $x$ whose *knots* are the points where the operand inside a ReLU activation changes sign, namely $e_i = b_i/a_i$, $a_i \ne 0$, $i = 1, \ldots, m$. Alternatively, we can visualize the canonical parameters $\boldsymbol{w} = \pi(\boldsymbol{z})$ in $\mathbb{R}^2$, by plotting a neuron $(r_i, \theta_i)$ as a particle with coordinates

$$(u_i, v_i) = (|r_i| \cos(\theta_i), |r_i| \sin(\theta_i)), \tag{6}$$

and coloring each particle according to $\epsilon_i = \text{sign}(r_i)$ (Figure 1, right). In this visualization, each training sample point $x_j$ can be represented as the line $ux_j + v = 0$, which identifies the half-space of neurons that are active at $x_j$. The collection of such lines for all samples partitions the plane into *activation regions*, where neurons have a fixed *activation pattern* on the training data.

# 3   Training Dynamics

Our goal is to solve (1) using the gradient flow (2) of the loss $L(\boldsymbol{z})$[1]. We begin in Section 3.1 by investigating the gradient dynamics in the canonical parameterization:

$$\boldsymbol{w}(0) = \boldsymbol{w}_0, \qquad \boldsymbol{w}'(t) = -\nabla \tilde{L}(\boldsymbol{w}(t)). \tag{7}$$

While the relationship between flows (2) and (7) is nonlinear, we argue in Section 3.2 that these are related by a simple change of metric.

## 3.1   Dynamics in the Canonical Parameters

We assume that the canonical parameters $(r_i, \theta_i)$ are initialized i.i.d. from some base distribution $\mu(r, \theta)$. The function $\tilde{f}_{\boldsymbol{w}}$ is well-defined pointwise as $m \to \infty$, by the law of large numbers. Following the mean-field formulation of single-hidden layer networks [22, 8, 26], we express the function as an expectation with respect to the probability measure over the cylinder $\mathcal{D} = \mathbb{R} \times S^1$:

$$\tilde{f}_{\boldsymbol{w}}(x) = \int_{\mathcal{D}} \varphi(w; x) \mu^{(m)}(\mathrm{d}w) \,,$$

where $\varphi(w; x) := r_i \langle \tilde{x}, d(\theta_i) \rangle_+$ and $\mu^{(m)}(w) = \frac{1}{m} \sum_{i=1}^m \delta_{w_i}(w)$ is the empirical measure determined by the $m$ particles $w_i$, $i = 1 \dots m$. The least squares loss in this case becomes

$$\begin{aligned}
\tilde{L}(\boldsymbol{w}) &= \frac{1}{2} \|\tilde{f}_{\boldsymbol{w}} - y\|_{\mathcal{X}}^2 \\
&= C_y - \frac{1}{m} \sum_{i=1}^m \langle \varphi_{w_i}, y \rangle_{\mathcal{X}} + \frac{1}{2m^2} \sum_{i, i'=1}^m \langle \varphi_{w_i}, \varphi_{w_{i'}} \rangle_{\mathcal{X}} \,,
\end{aligned}$$

where $\langle f, g \rangle_{\mathcal{X}} := \sum_{j=1}^s f(x_j) g(x_j)$ is the empirical dot-product. This loss may be interpreted as the Hamiltonian of a system of $m$-interacting particles, under external field $F$ and interaction kernel $K$ defined respectively by $F(w) := \langle \varphi_w, y \rangle_{\mathcal{X}}$, $K(w, w') := \langle \varphi_w, \varphi_{w'} \rangle_{\mathcal{X}}$. We may also express this Hamiltonian in terms of the empirical measure, by abusing notation

$$\tilde{L}(\mu^{(m)}) = C_y - \int_{\mathcal{D}} F(w) \mu^{(m)}(\mathrm{d}w) + \frac{1}{2} \iint_{\mathcal{D}^2} K(w, w') \mu^{(m)}(\mathrm{d}w) \mu^{(m)}(\mathrm{d}w') \,.$$

A direct calculation shows that the gradient $\nabla_{w_i} \tilde{L}(\boldsymbol{w})$ can be written as

$$\frac{m}{2} \nabla_{w_i} \tilde{L}(\boldsymbol{w}) = \nabla_w V(w_i; \mu^{(m)}) \,,$$

where $V$ is the potential function $V(w; \mu) := -F(w) + \int_{\mathcal{D}} K(w, w') \mu(\mathrm{d}w')$. The gradient flow in the space of parameters $\boldsymbol{w}$ can now be interpreted in Eulerian terms as a gradient flow in the space of measures over $\mathcal{D}$, by using the notion of Wasserstein gradient flows [22, 8, 26]. Indeed, particles evolve in $\mathcal{D}$ by "feeling" a velocity field $\nabla V$ defined in $\mathcal{D}$. This formalism allows us now to describe the dynamics independently of the number of neurons $m$, by replacing the empirical measure $\mu^{(m)}$ by any generic probability measure $\mu$ in $\mathcal{D}$. The evolution of a measure under a generic time-varying vector field is given by the so-called continuity equation:[2]

$$\partial_t \mu_t = \mathrm{div}(\nabla V \mu_t) \,. \tag{8}$$

The global convergence of this PDE for interaction kernels arising from single-hidden layer neural networks has been established under mild assumptions in [22, 8, 25]. Although the conditions for global convergence hold in the mean field limit $m \to \infty$, a propagation-of-chaos argument from statistical mechanics gives Central Limit Theorems for the behavior of finite-particle systems as fluctuations of order $1/\sqrt{m}$ around the mean-field solution; see [26, 25] for further details.

The dynamics in $\mathcal{D}$ are thus described by the velocity field $\nabla V(w; \mu_t)$, which depends on the current state of the system through the measure $\mu_t(w)$, describing the probability of encountering a particle

at position $w$ at time $t$. We emphasize that equation (8) is valid for any measure, including the empirical measure $\mu^{(m)}$, and is therefore an exact model for the dynamics in both the finite-particle and infinite-particle regime. Let us now describe its specific form in the case of the empirical loss given above.

Assume without loss of generality that the data points $x_j \in \mathbb{R}$, $j \le s$ satisfy $x_j \le x_{j'}$ whenever $j < j'$. Denote

$$\mathcal{C}_j := \{j'; j' \le j\} \text{ for } j = 1 \ldots s, \qquad \mathcal{C}_{s+j} := \{j'; j' > j\}, \text{ for } j = 1 \ldots s - 1 \,.$$

We observe that for each $j$, two angles $\alpha_j^{\pm} = \arctan(x_j) \pm \pi/2$ partition the circle $S^1$ into $2s - 1$ regions $\mathcal{A}_k$ (visualized as the colored regions in Figure 1), which are in one-to-one correspondence with the sets $\mathcal{C}_k$, in the sense that $\theta \in \mathcal{A}_k$ if and only if $\{j; \langle \tilde{x}_j, d(\theta) \rangle \ge 0\} = \mathcal{C}_k$. We also denote by $\mathcal{B}_j$ the half-circle where $\langle \tilde{x}_j, \theta \rangle \ge 0$. Let $t(\theta)$ be the tangent vector of $S^1$ at $\theta$ (so $t(\theta) = d(\theta)^{\perp}$) and $w = (r, \theta)$, where we suppose $\theta \in \mathcal{A}_k$. A straightforward calculation (see Appendix B) shows that the angular and radial components of $\nabla V(w; \mu_t)$ are given by

$$\nabla_{\theta} V(w; \mu_t) = r \left\langle \sum_{j \in \mathcal{C}_k} \rho_j(t) \tilde{x}_j, t(\theta) \right\rangle, \qquad \nabla_r V(w; \mu_t) = \left\langle \sum_{j \in \mathcal{C}_k} \rho_j(t) \tilde{x}_j, d(\theta) \right\rangle, \quad (9)$$

where $\rho_j(t) = \int_{\mathbb{R} \times \mathcal{B}_j} r \langle \tilde{x}_j, \theta \rangle \mu_t(\mathrm{d}r, \mathrm{d}\theta) - y_j$ is the residual at point $x_j$ at time $t$. These expressions show that the dynamics are entirely controlled by the $s$-dimensional vector of residuals $\boldsymbol{\rho}(t) = (\rho_1(t), \ldots \rho_s(t))$, and that the velocity field is *piece-wise linear* on each cylindrical region $\mathbb{R} \times \mathcal{A}_k$ (e.g. Figure 9 in Appendix D). Under the assumptions that ensure global convergence of (8), we have $\lim_{t \to \infty} \tilde{L}(\mu_t) = 0$, and therefore $\|\boldsymbol{\rho}(t)\| \to 0$. The oscillations of $\boldsymbol{\rho}(t)$ as it converges to zero determine the relative orientation of the flow within each region. The exact dynamics for the vector of residuals are given by the following proposition, proved in Appendix B:

**Proposition 1.** *For each $j$,*

$$\dot{\rho}_j(t) = -\tilde{x}_j^{\top} \sum_{k; \mathcal{A}_k \subset \mathcal{B}_j} \Sigma_k(t) \left( \sum_{j' \in \mathcal{C}_k} \rho_{j'}(t) \tilde{x}_{j'} \right), \quad (10)$$

*where $\Sigma_k(t) = \int_{\mathbb{R} \times \mathcal{A}_k} \left( r^2 t(\theta) \, t(\theta)^{\top} + d(\theta) \, d(\theta)^{\top} \right) \mu_t(\mathrm{d}r, \mathrm{d}\theta)$ tracks the covariance of the measure along each cylindrical region.*

Equation (10) defines a system of ODEs for the residuals $\boldsymbol{\rho}(t)$, but its coefficients are time-varying, and behave roughly as quadratic terms in $\boldsymbol{\rho}(t)$ (since they are second-order moments of the measure whereas the residuals are first-order moments). It may be possible to obtain asymptotic control of the oscillations $\boldsymbol{\rho}(t)$ by applying Duhamel's principle, but this is left for future work.

Now let $w = (r, \theta)$ with $\theta$ at a boundary of two regions $\mathcal{A}_k$, $\mathcal{A}_{k+1}$. The velocity field is modified at the transition by

$$\nabla V(w)|_{\mathcal{A}_k} - \nabla V(w)|_{\mathcal{A}_{k+1}} = \rho_{j*}(t) \begin{pmatrix} r \langle \tilde{x}_{j*}, t(\theta) \rangle \\ \langle \tilde{x}_{j*}, \theta \rangle \end{pmatrix},$$

where $j_*$ is such that $\langle \tilde{x}_{j*}, d(\theta) \rangle = 0$, since $\theta$ is at the boundary of $\mathcal{A}_k$. It follows that the only discontinuity is in the angular direction, of magnitude $|r \rho_{j*}(t)| \|\tilde{x}_{j*}\|$. In particular, an interesting phenomenon arises when the angular components of $\nabla V(w)|_{\mathcal{A}_k}$ and $\nabla V(w)|_{\mathcal{A}_{k+1}}$ have opposite signs, corresponding to an "attractor" or "repulsor" that attracts/repels particles along the direction given by $\tilde{x}_{j*}$ (See Figure 9 in Appendix D). Writing $s_k = \left\langle \sum_{j \in \mathcal{C}_k} \rho_j(t) \tilde{x}_j, t(\theta) \right\rangle$, we deduce from (9) that this occurs when $|s_k| < |\rho_{j*}(t)| \|\tilde{x}_{j*}\|$ and $\text{sign}(s_k) \ne \text{sign}(\rho_{j*}(t))$. We expand this condition in the following Lemma.

**Lemma 2.** *A data point $x_k$ is an attractor/repulsor if and only if*

$$\sum_{i=1}^{k-1} \rho_i \rho_k \langle \tilde{x}_i, \tilde{x}_k \rangle > -\rho_k^2 \|\tilde{x}_k\|^2, \text{ or } \sum_{i=k+1}^{s} \rho_i \rho_k \langle \tilde{x}_i, \tilde{x}_k \rangle > -\rho_k^2 \|\tilde{x}_k\|^2.$$

In words, mass will concentrate towards input points where the residual is currently large and of opposite sign from a weighted average of neighboring residuals. This is in stark contrast with the kernel dynamics (Section 3.3), where there is no adaptation to the input data points. We point out that this qualitative behavior has been established in [21] under appropriate initial conditions, sufficiently close to zero, in line with our mean-field analysis. We also refer to Section B.2 of the Appendix, where we describe the adaptive regime when the objective is augmented with TV regularization.

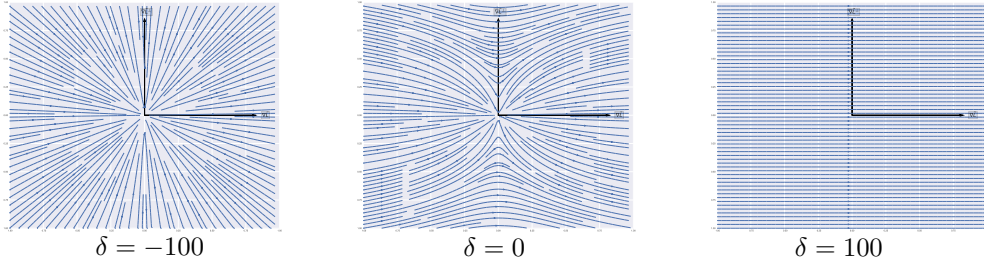

$$\delta = -100 \qquad\qquad \delta = 0 \qquad\qquad \delta = 100$$

Figure 2: The value of $\delta$ interpolates between different kinds of local trajectories of neurons. The plots are in the coordinate frame $(\nabla \tilde{L}, \nabla \tilde{L}^{\perp})$. Left: the neurons move radially towards and away from the origin. Middle: the trajectories containing both radial and tangential components. Right: the trajectories are parallel to the gradient $\nabla \tilde{L}$.

## 3.2 Dynamics in the Full Parameters

The dynamics of gradient flow (2) are different from the dynamics of the gradient flow (7). For the gradient flow in canonical parameters we have observed adaptive learning behavior under the assumption of iid distribution of parameter initialization. The full set of parameters $\boldsymbol{z} = (\boldsymbol{a}, \boldsymbol{b}, \boldsymbol{c})$, may exhibit both kernel and adaptive behavior depending on the initialization. To characterize this behavior we rely on the following lemma.

**Lemma 3.** *If $\boldsymbol{z}(t) = (\boldsymbol{a}(t), \boldsymbol{b}(t), \boldsymbol{c}(t))$ is a solution of the gradient flow* (2)*, then the quantities*

$$\boldsymbol{\delta} = (c_i(t)^2 - a_i(t)^2 - b_i(t)^2)_{i=1}^m \tag{11}$$

*remain constant for all $t$. In particular, given a reduced neuron $(r_i, \theta_i)$, we can uniquely recover the original neuron parameters $(a_i, b_i, c_i)$ from $\delta_i$ computed from the initialization.*

Lemma 3 allows us to analyze how canonical parameters evolve under *full* gradient flow in $(\boldsymbol{a}, \boldsymbol{b}, \boldsymbol{c})$. Overall, the behavior is qualitatively the same, except it is in addition dependent on the relative scale of redundant parameters.

**Proposition 4.** *Let $\boldsymbol{z}(t)$ be a solution of the gradient flow* (2) *of $L(\boldsymbol{z})$, and let $\boldsymbol{\delta} = (\delta_i) \in \mathbb{R}^m$ be the vector of invariants* (11)*, which depend only on the initialization $\boldsymbol{z}(0)$. If $\boldsymbol{w}(t) = (\boldsymbol{r}(t), \boldsymbol{\theta}(t))$ is curve of canonical parameters corresponding to $\boldsymbol{z}(t)$, then we have that*

$$\dot{\boldsymbol{w}}_i(t) = \boldsymbol{P}_i \cdot \nabla_{\boldsymbol{w}_i} \tilde{L}(\boldsymbol{w}), \quad i = 1, \ldots, m, \tag{12}$$

*where*

$$\boldsymbol{P}_i = \begin{bmatrix} \frac{m^2}{\alpha(m)^2}(a_i^2 + b_i^2 + c_i^2) & 0 \\ 0 & \frac{1}{a_i^2 + b_i^2} \end{bmatrix}. \tag{13}$$

With respect to rescaled differentials $d\tau = r d\theta$, corresponding to representing the flow locally in a Cartesian system aligned with the radial direction (pointing away from $\boldsymbol{z} = \boldsymbol{0}$) and its perpendicular, the flow can be written as

$$\begin{bmatrix} dr_i \\ d\tau_i \end{bmatrix} = \begin{bmatrix} \frac{m^2}{\alpha(m)^2}(a_i^2 + b_i^2 + c_i^2) & 0 \\ 0 & c_i^2 \end{bmatrix} \cdot \begin{bmatrix} \nabla_{r_i} \tilde{L}(\boldsymbol{w}) dt \\ \nabla_{\tau_i} \tilde{L}(\boldsymbol{w}) dt \end{bmatrix}, \quad i = 1, \ldots, m, \tag{14}$$

From these equations, one can see that if $c_i^2 \ll a_i^2 + b_i^2$ for all $i$ (*i.e.*, $\delta_i \ll 0$), then radial motion will dominate. In other words, initializing the first layer with significantly larger values than the second leads to kernel learning. On the other hand, if $c_i^2 \gg a_i^2 + b_i^2$, then a solution of the gradient flow (2) will follow the same trajectory as for the reduced gradient flow (7). Also, if $\alpha(m) = o(m)$, the radial component will dominate as $m$ increases. Figure 2 shows the trajectories corresponding to different values of $\delta_i$ for each neuron, with $\alpha(m) = m$. The extreme cases of $\delta = -\infty$ and $\delta = +\infty$ correspond to the "kernel" and "adaptive" regimes, respectively. Note that as $\delta$ increases, the neurons cluster at sample points, as explained in our analysis in Section 3.1, and in accordance to [21].

### 3.3 Kernel Dynamics

We now consider the dynamics in the special case where $\delta \ll 0$, and we consider $m \to \infty$. To obtain the kernel regime in this case, it is sufficient to consider a normalization $\alpha(m) = o(m)$. In particular, when $\alpha(m) = 1$, as shown in the previous section, the parameters $\boldsymbol{a}$ and $\boldsymbol{b}$ remain mostly fixed and the parameters $\boldsymbol{c}$ change throughout training, corresponding to the so-called random-features (RF) kernel of Rahimi and Recht [24].

In the limit case where $\boldsymbol{a}$ and $\boldsymbol{b}$ are completely fixed to their initial values, if we choose $\boldsymbol{c}$ close to the zero vector, then the least squares problem (1) solved using gradient flow, is equivalent to the minimal-norm constraint problem solution:

$$
\begin{aligned}
&\text{minimize } \|\boldsymbol{c}\|^2 \\
&\text{subject to } f_{\boldsymbol{z}}(x_i) = y_i, \qquad i = 1, \dots, s.
\end{aligned}
\tag{15}
$$

Given an initial distribution $\mu_0$ over the domain $\mathcal{D}_a \times \mathcal{D}_b$ of parameters $a$ and $b$, the random-feature (RF) kernel associated with (15) is given by

$$
K_{\mathrm{RF}}(x, x') = \int_{\mathcal{D}_a \times \mathcal{D}_b} [xa - b]_+ \cdot [x'a - b]_+ \mu_0(\mathrm{d}a, \mathrm{d}b) .
\tag{16}
$$

The solution of (15) can now be written in terms of this RF kernel using the representer theorem $\tilde{f}_{\boldsymbol{z}}(x) = \sum_{j=1}^s \alpha_j K_{\mathrm{RF}}(x_j, x)$, where $\alpha$ is a vector of minimal RKHS norm that fulfills the interpolation constraints. Under appropriate assumptions, the solution to (15) is a *cubic spline*.

**Theorem 5.** *Assume the measure $\mu_0$ has finite second moment $\sigma_{\mu_0}^2 := \mathbb{E}_{(a,b)\sim\mu_0}(a^2 + b^2) < \infty$. Let $\mu_0(a, b) = q(a)\mu_a(b)$ be the decomposition in terms of marginal and conditional, and assume $\mu_a$ is bounded for each $a$. Define $\nu(u) = \int |a|\mu_a(ab)\mathrm{d}q(a)$. Then the solution (15) solves*

$$
\min_f \quad \|f\|_{\mathrm{RF}}^2 := \int_\Omega \frac{|f''(u)|^2}{\nu(u)}\mathrm{d}u \quad s.t. \quad f(x_i) = y_i \, , i = 1 \dots s \, ,
\tag{17}
$$

*where $\Omega := supp(\nu)$. Moreover, if $\mu_0$ is such that $\mu_0(a, b) = q(a)\mathbf{1}(b \in I_a)$, where $I_a \subset \mathbb{R}$ is an arbitrary interval, then (15) will be a cubic spline.*

Notice that the assumptions on $\mu_0$ to obtain an exact cubic spline kernel impose that if $A, B$ is a random vector distributed according to $\mu_0$, then $B|A$ is uniform over an arbitrary interval $I_A$ that can depend upon $A$. The proof illustrates that one may generalize the interval $I_A$ by any countable union of intervals. In particular, independent uniform initialization yields cubic splines, but radial distributions, such as $A, B$ being jointly Gaussian, do not (see Section A.3 in the Appendix). We remark that machine learning packages such as PyTorch use a uniform distribution for linear layer parameter initialization by default. We verify that indeed, solutions to (1) converge to cubic splines as $m$ grows in Figure 3. We also point out that in Kernel Learning, early termination of gradient flow acts as a regularizer favoring smooth, non-interpolatory solutions (see [19]).

The analysis and comparison of these kernels has recently been addressed in [5, 14] in the general high-dimensional setting, by describing its spectral decay in terms of spherical harmonics. Our results complement them in the particular one-dimensional setting thanks to the explicit functional form of the resulting RKHS norms. Additionally, Savarese et al. [28] study the functional form of the minimization in the variation norm, leading to a penalty of the form $\int |f''(u)|\mathrm{d}u$. We have instead $L^2$ norms (RKHS) in the kernel regime. The $L^2$ norms do not provide any adaptivity as opposed to the $L^1$ norm [3]. An interesting question is to precisely describe the transition between these two regimes as a function of the initialization.

**Numerical Experiments.** For our numerical experiments, we use gradient descent with the parameterization (1) and $\alpha(m) = \sqrt{m}$, appropriately scaling the weights $\boldsymbol{a}, \boldsymbol{b}, \boldsymbol{c}$ to achieve different dynamical behaviors. We also refer to Section D in the Appendix for additional experiments.

*Cubic Splines.* We show in Figure 3 that when $-\delta \ll r^2$ (*i.e.*, in the kernel regime), and as the number of neurons grows, the network function $f_{\boldsymbol{z}}$ converges to a cubic spline. For this experiment, we used 10 points sampled from a square wave, and trained only the parameters $\boldsymbol{c}$ (*i.e.*, $\delta_i = \infty$).

*Network Dynamics as a Function of $\delta$.* We show in Figure 4 that as we vary $\delta$, the network function goes from being smooth and non-adaptive in the kernel regime ($\delta = -\infty$, *i.e.* training only the parameter $\boldsymbol{c}$) to very adaptive ($\delta = \infty$, *i.e.* training only the parameters $\boldsymbol{a}, \boldsymbol{b}$). Note that as $\delta$ increases, clusters of knots emerge at the sample positions (collinear points in the $uv$ diagrams).

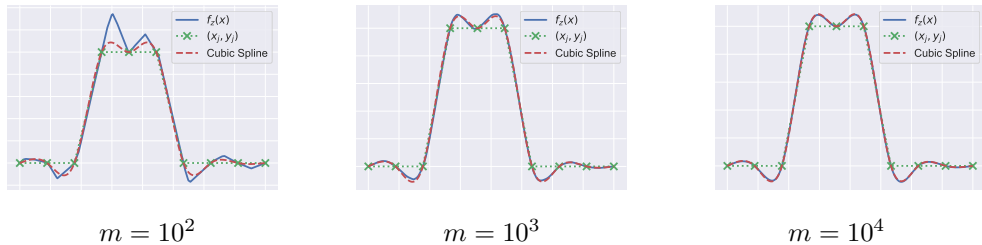

$$m = 10^2 \qquad\qquad m = 10^3 \qquad\qquad m = 10^4$$

Figure 3: A cubic spline with vanishing second derivative at its endpoints (blue line) is approximated by a neural network ($\delta = -100$) while varying the number $m$ of neurons.

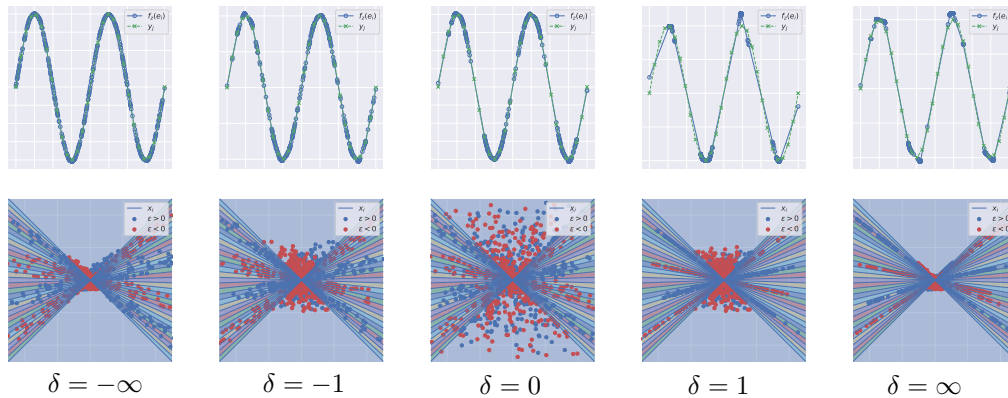

$$\delta = -\infty \qquad \delta = -1 \qquad \delta = 0 \qquad \delta = 1 \qquad \delta = \infty$$

Figure 4: Comparison of fitting the network function to a sinusoid as $\delta$ varies (10000 epochs).

## 4 Concluding Remarks

We have studied the implicit bias of gradient descent in the approximation of univariate functions with single-hidden layer ReLU networks. Despite being an extremely simplified learning setup, it provides a clear illustration that such implicit bias can be drastically different depending on how the neural architecture is parameterized, normalized, or even initialized. Building on recent theoretical work that studies neural networks in the overparameterized regime, we show how the model can behave either as a 'classic' cubic spline interpolation kernel, or as an adaptive interpolation method, where neurons concentrate on sample points where the approximation most needs them. Moreover, in the one-dimensional case, we complement existing works [29] to reveal a transition between these two extreme training regimes, which roughly correspond to $W^{1,2}$ and $W^{2,2}$ Sobolev spaces respectively. Although in our univariate setup there is no clear advantage of one functional space over the other, our full description of the dynamics may prove useful in the high-dimensional regime, where the curse of dimensionality affects Hilbert spaces defined by kernels [3]. We believe that the analysis of the PDE resulting from the mean-field regime (where adaptation occurs) in the low-dimensional setting will be useful to embark in the analysis of the high-dimensional counterpart. We note however that naively extending our analysis to high-dimensions would result in an exponential increase in the number of regions that define our piecewise linear flow, thus we anticipate that new tools might be needed. Moreover, the interpretation of ReLU features in terms of Green's functions (as first pointed out in [29]) does not directly carry over to higher dimensions. Lastly, another important limitation of the mean-field analysis is that it cannot be easily adapted to deep neural network architectures, since neurons are no longer exchangeable as in the many-particle system described above.

**Acknowledgements:** This work was partially supported by the Alfred P. Sloan Foundation, NSF RI-1816753, NSF CAREER CIF 1845360, Samsung Electronics, the NSF CAREER award 1652515, the NSF grant IIS-1320635, the NSF grant DMS-1436591, the NSF grant DMS-1835712, the SNSF grant P2TIP2_175859, the Moore-Sloan Data Science Environment, the DARPA D3M program, NVIDIA, Labex DigiCosme, DOA W911NF-17-1-0438, a gift from Adobe Research, and a gift from nTopology. Any opinions, findings, and conclusions or recommendations expressed in this material are those of the authors and do not necessarily reflect the views of DARPA.

## Footnotes

[1]To be precise, we should replace the gradient $\nabla L(\boldsymbol{z})$ with the *Clarke subdifferential* $\partial L(\boldsymbol{z})$ [9], since $L(\boldsymbol{z})$ is only piecewise smooth. At generic smooth points $\boldsymbol{z}$, the subdifferential coincides with the gradient $\partial L(\boldsymbol{z}(t)) = \{\nabla L(\boldsymbol{z})\}$.

[2]Understood in the weak sense, *i.e.*, $\partial_t \left( \int_{\mathcal{D}} \phi(w) \mu_t(\mathrm{d}w) \right) = -\int \langle \nabla \phi(w), \nabla V(w; \mu_t) \rangle \mu_t(\mathrm{d}w)$, $\forall \phi \in C_c^1(\mathcal{D})$ continuously differentiable and with compact support.

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
