[Supplementary Material]

# Appendix

 ## A  The Functional Space of ReLU Networks

We consider the class of 1D shallow ReLU functions with exactly $m$ neurons:

$$\mathcal{F}_m = \{f_{\boldsymbol{\theta}} : \mathbb{R} \to \mathbb{R}\}, \qquad f_{\boldsymbol{\theta}}(x) = \sum_{i=1}^{m} c_i[a_i x + b_i]_+. \tag{23}$$

It is easy to see that $\mathcal{F}_m$ is a subset of the set $CPL_m$ of *continuous piecewise-linear maps* with at most $m$ knots.

**Proposition 5.** *The space $\mathcal{F}_m$ is a strict subset of $CPL_m$, but it contains $CPL_{m-2}$.*

*Proof.* The fact that $\mathcal{F}_m \subsetneq CPL_m$ can be argued by observing that $\mathcal{F}_m$ has $2m$ degrees of freedom, whereas $CPL_m$ has $2m + 2$ degrees of freedom. More precisely, we can partition the parameter space of $f_{\boldsymbol{\theta}}$ based on the signs $\sigma_i = \text{sign}(a_i)$. Within each region, we may write

$$f_{\boldsymbol{\theta}}(x) \sum_{i=1}^{m} c_i[a_i(x - e_i)]_+ = \sum_{i=1}^{m} u_i[\sigma_i(x - e_i)]_+, \qquad u_i = |a_i| c_i. \tag{24}$$

We see that interpolatory constraints on $f_{\boldsymbol{\theta}}(x)$ correspond to linear conditions on $u_i$. In particular, there are a finite number of functions in $\mathcal{F}_m$ with fixed assigned (generic) values at the knots $e_1 < \ldots < e_m$. This contrasts with $CPL_m$, where the slope in the intervals $[-\infty, e_1]$ and $[e_m, \infty]$ may be chosen arbitrarily. Finally, by setting $e_1 = e_2$ and $e_{m-1} = e_m$ we can control the two remaining slopes, so we recover that $CPL_{m-2} \subset \mathcal{F}_m$. $\qquad\qquad\square$

 ## B  Spline Kernels

**Proposition 6.** *If either (i) $a(s)$ is identically 1 or (ii) measures of supports $a_+(s)$ and $a_-(s)$ as $m \to \infty$ on any subinterval of $[k_0, k_1]$ have the same expectation, the kernel $K(x, x')$ defined as*

$$K(x, x') = \int a_+(s)^2 [x - s]_+ [x' - s]_+ ds + \int a_-(s)^2 [s - x]_+ [s - x']_+ ds$$

*is a piecewise cubic polynomial in $x$ and $x'$. In particular, in the overparameterized setting, a function $\hat{f}(x) = \sum_{i=1}^{s} \alpha_i K_r(x, x_i)$ that interpolates the samples $x_1, \ldots, x_s$ will be* cubic spline, *with knots at these samples.*

*Proof.* In case (i), assuming that $x < x'$, we have that

$$
\begin{aligned}
K_1(x, x') = \int_{k_0}^{k_1} [x - s]_+ [x' - s]_+ ds &= \int_{k_0}^{x} (x - s)(x' - s) ds \\
&= \left[ \frac{1}{3} s^3 - \frac{1}{2} s^2 (x + x') + s x x' \right]_{k_0}^{x} \\
&= -\frac{1}{6}(2\,k_0 + x - 3\,x')(k_0 - x)^2
\end{aligned} \tag{25}
$$

In the second case, each of the finite sums approximating the integrals remains constant, and contains half of the indices, which are distributed densely both for $a_+$ and $a_-$. The limit integrals are

$$
\begin{aligned}
K_2(x, x') = \frac{1}{2} \int_{k_0}^{k_1} [x - s]_+ [x' - s]_+ ds &+ \frac{1}{2} \int_{k_0}^{k_1} [-x + s]_+ [-x' + s]_+ ds \\
&= \frac{1}{2} \int_{k_0}^{x} (x - s)(x' - s) ds + \frac{1}{2} \int_{x'}^{k_1} (s - x)(s - x') ds. \\
&= -\frac{1}{12}(2\,k_0 + x - 3\,x')(k_0 - x)^2 + \frac{1}{12}(2\,k_1 - 3\,x + x')(k_1 - x')^2.
\end{aligned} \tag{26}
$$

326 Both $K_1(x, x')$ and $K_2(x, x')$ are piecewise cubic and $C^2$ in both arguments. This immediately
327 implies that the solution to the least squares problem $\hat{f}(x) = \sum_{i=1}^{s} \alpha_i K_t(x, x_i)$ $(t = 1, 2)$ is a *cubic*
328 *spline* interpolating the samples $x_1, \ldots, x_s$. □

329 Finally the following simple fact shows that the coefficient function $c(x)$ effectively corresponds to
330 the second derivative (or *linearized curvature*) $f_z''(x)$ of $f_z(x)$.

331 **Lemma 7.** *Consider a function of the form*

$$f(x) = \int_{k_0}^{k_1} c^+(s)[x-s]_+ ds + \int_{k_0}^{k_1} c^-(s)[s-x]_+ ds, \qquad k_0 \le x \le k_1. \tag{27}$$

332 *Then we have that*

$$f'(x) = \int_a^x c^+(\theta)d\theta - \int_x^b c^-(\theta)d\theta, \qquad f''(x) = c^+(x) + c^-(x). \tag{28}$$

333 *Proof.* This follows by observing that

$$f(x) = \int_{k_0}^x c^+(s)(x-s)d\theta + \int_x^{k_1} c^-(s)(s-x)ds, \tag{29}$$

334 and applying the Leibniz integral differentiation rule. □

335 # C  Mean Field Computations

336 Making use of notation introduced in Section 3.1, we have that if $w = (\hat{r}, \theta)$, with $\theta \in \mathcal{A}_k$, then

$$
\begin{aligned}
\nabla_\theta V(w; \mu_t) &= -\nabla_\theta F(w) + \int_{\mathcal{D}} \nabla_\theta K(w, w') \mu_t(dw') \\
&= -\hat{r}\left( \sum_{j \in \mathcal{C}_k} y_j \langle \tilde{x}_j, t(\theta) \rangle - \int_{\mathcal{D}} r' \sum_{j \in \mathcal{C}_k} \langle \tilde{x}_j, t(\theta) \rangle \langle \tilde{x}_j, d(\theta') \rangle_+ \mu_t(d\hat{r}', d\theta') \right) \\
&= -\hat{r} \sum_{j \in \mathcal{C}_k} \langle \tilde{x}_j, t(\theta) \rangle \left( y_j - \int_{\mathbb{R} \times \mathcal{B}_j} \hat{r}' \langle \tilde{x}_j, d(\theta') \rangle \mu_t(d\hat{r}', d\theta') \right) \\
&= \hat{r} \left\langle \sum_{j \in \mathcal{C}_k} \rho_j(t) \tilde{x}_j, t(\theta) \right\rangle ,
\end{aligned}
\tag{30}
$$

337 where $\rho_j(t) = f_{\mu_t}(x_j) - y_j = \int_{\mathbb{R} \times \mathcal{B}_j} c \langle \tilde{x}_j, \theta \rangle \mu_t(d\hat{r}, d\theta) - y_j$ is the residual at point $x_j$ at time $t$.
338 Similarly, the field in the direction of the charges is given by

$$\nabla_{\hat{r}} V(w; \mu_t) = \left\langle \sum_{j \in \mathcal{C}_k} \rho_j(t) \tilde{x}_j, \theta \right\rangle . \tag{31}$$

339 We also observe that for each $j$,

$$
\begin{aligned}
\dot{\rho}_j(t) &= \partial_t f_{\mu_t}(x_j) = \partial_t \left( \int_{\mathcal{D}} \varphi(w; x_j) \mu_t(dw) \right) \\
&= -\int_{\mathcal{D}} \langle \nabla_w \varphi(w; x_j), \nabla V(w; \mu_t) \rangle \mu_t(dw) \\
&= -\int_{\mathcal{D}} \left( \nabla_\theta \varphi(w; x_j) \cdot \nabla_\theta V(w; \mu_t) + \nabla_r \varphi(w; x_j) \cdot \nabla_r V(w; \mu_t) \right) \mu_t(dw) \\
&= -\sum_{k; \mathcal{A}_k \subset \mathcal{B}_j} \int_{\mathbb{R} \times \mathcal{A}_k} \left( r^2 \tilde{x}_j^\top (t(\theta)t(\theta)^\top)(\sum_{j' \in \mathcal{C}_k} \rho_{j'}(t)\tilde{x}_{j'}) + \tilde{x}_j^\top (\theta\theta^\top)(\sum_{j' \in \mathcal{C}_k} \rho_{j'}(t)\tilde{x}_{j'}) \right) \mu_t(dw) \\
&= -\tilde{x}_j^\top \sum_{k; \mathcal{A}_k \subset \mathcal{B}_j} \Sigma_k(t)(\sum_{j' \in \mathcal{C}_k} \rho_{j'}(t)\tilde{x}_{j'}) ,
\end{aligned}
\tag{32}
$$

where

$$\Sigma_k(t) = \int_{\mathbb{R} \times \mathcal{A}_k} \left( r^2 t(\theta) \, t(\theta)^\top + \theta \, \theta^\top \right) \mu_t(dr, d\theta)$$

tracks the covariance of the measure along each cylindrical region. Equation (32) defines a system of ODEs for the residuals $\boldsymbol{\rho}(t)$, but its coefficients are time-varying, and behave roughly as quadratic terms in $\boldsymbol{\rho}(t)$ (since they are second-order moments of the measure whereas the residuals are first-order moments). It may be possible to obtain asymptotic control of the oscillations $\boldsymbol{\rho}(t)$ by applying Duhamel's principle.

# D  Changing Metric in the Dynamics

**Lemma 8.** *If* $\boldsymbol{z}(t) = (\boldsymbol{a}(t), \boldsymbol{b}(t), \boldsymbol{c}(t))$ *is a solution of the gradient flow* (5)*, then the quantities*

$$\boldsymbol{\delta} = (\delta_i = c_i(t)^2 - a_i(t)^2 - b_i(t)^2)_{i=1}^m \tag{33}$$

*remain constant for all t. In particular, given a reduced neuron* $(r_i, \theta_i)$*, we can uniquely recover the original neuron* $(a_i, b_i, c_i)$*, since*

$$c_i^2 = \frac{\delta_i + \sqrt{\delta_i^2 + 4r_i^2}}{2}. \tag{34}$$

*Proof.* The gradient equations of the loss $L(\boldsymbol{z})$ can be written as

$$\nabla_{a_i} L(\boldsymbol{z}) = c_i \sum_{j=1}^s \mathbb{1}[a_i x_j + b_i \geq 0] x_j r_j,$$

$$\nabla_{b_i} L(\boldsymbol{z}) = c_i \sum_{j=1}^s \mathbb{1}[a_i x_j + b_i \geq 0] r_j, \tag{35}$$

$$\nabla_{c_i} L(\boldsymbol{z}) = \sum_{j=1}^s \mathbb{1}[a_i x_j + b_i \geq 0](a_i x_j + b_i) r_j.$$

From these expressions we see that

$$\begin{aligned}
\dot{\delta}_i &= 2c_i \dot{c}_i - 2a_i \dot{a}_i - 2b_i \dot{b}_i \\
&= 2c_i \nabla_{c_i} L(\boldsymbol{z}) - 2a_i \nabla_{a_i} L(\boldsymbol{z}) - 2b_i \nabla_{b_i} L(\boldsymbol{z}) \\
&= 0.
\end{aligned}$$

Using $r_i^2 = c_i \sqrt{a_i^2 + b_i^2}$, we see that $c_i^2 - \frac{r_i^2}{c_i^2} = \delta_i$ implies $c_i^4 - \delta_i c_i^2 - r^2 = 0$, and thus (34). $\square$

**Theorem 9.** *Let* $\boldsymbol{z}(t)$ *be a solution gradient flow* (5) *of* $L(\boldsymbol{z})$*, and let* $\boldsymbol{\delta} = (\delta_i) \in \mathbb{R}^m$ *be the vector of invariants* (15)*, which depend only on the initialization* $\boldsymbol{z}(0)$*. If* $\boldsymbol{w}(t) = (\boldsymbol{r}(t), \boldsymbol{\theta}(t))$ *is curve of reduced parameters corresponding to* $\boldsymbol{z}(t)$*, then we have that*

$$\dot{\boldsymbol{w}}_i(t) = \boldsymbol{P}_i \cdot \nabla_{\boldsymbol{w}_i} \tilde{L}(\boldsymbol{w}), \quad i = 1, \ldots, m,$$

*where*

$$\boldsymbol{P}_{\delta_i}(r_i) = \begin{bmatrix} a_i^2 + b_i^2 + c_i^2 & 0 \\ 0 & \frac{1}{a_i^2 + b_i^2} \end{bmatrix} = \begin{bmatrix} \frac{r_i^2}{c(r_i)^2} + c(r_i)^2 & 0 \\ 0 & \frac{c(r_i)^2}{r_i^2} \end{bmatrix},$$

*and* $c(r_i)^2 = \frac{\delta_i + \sqrt{\delta_i^2 + 4r_i^2}}{2}$.

*Proof.* The Jacobian of the mapping $\boldsymbol{\pi}$ from parameters to reduced parameters is given by

$$Jac(\boldsymbol{\pi})(a_i, b_i, c_i) = \begin{bmatrix} \frac{ca}{\sqrt{a_i^2 + b_i^2}} & \frac{cb}{\sqrt{a_i^2 + b_i^2}} & \sqrt{a_i^2 + b_i^2} \\ -\frac{b}{a_i^2 + b_i^2} & \frac{a}{a_i^2 + b_i^2} & 0 \end{bmatrix}, \qquad i = 1, \ldots, m.$$

This implies that the *tangent kernel* $\boldsymbol{P}_{\delta_i}(r_i) = Jac(\boldsymbol{\pi}) Jac(\boldsymbol{\pi})^T$ is as in (17). We emphasize that the fact that this kernel can be written only as a function of $\boldsymbol{w}$ (and, in fact, only of $\boldsymbol{r}$) relies in essential manner on Lemma 2. $\square$

Figure 5: Evolution of 1000 neurons over 10000 epochs for $\delta = \pm\infty$ while fitting 10 points sampled from a square wave. Left: plotted network function after training. Middle: state of the network in $uv$ space after training. Right: training trajectories of 100 random neurons.

| Epoch 0 | Epoch $10^4$ | Epoch $10^4$ |
| $z = (a, b, c)$ | $z = (a, b, c)$ | $z = (10^3 a, 10^3 b, 10^{-3} c)$ |

Figure 6: Left: A network (green) with an initial set of parameters initial parameters $(a, b, c)$ is used to approximate a given function (blue). Scaling the initial parameters (right) leads to a very different fit (middle).

# E    Additional Numerical Experiments

In Figure 5, we plot the trajectories of neurons for $\delta = \pm\infty$ over 10000 epochs. We see that, if $\delta = -\infty$, the neurons move radially away from the origin and thus the knot positions do not change (top row). In stark contrast, if $\delta = \infty$, the neurons adapt to the input data, and the knots "stick" to input samples (bottom row).

We remark in Figure 6 that the same initial function can yield extremely different results depending on $\delta$.

We now show the effect of varying the number of neurons during training. In this example we fit 20 samples from a sine wave using 20, 200, and 2000 neurons respectively. In PyTorch, the default initializatioon is such that $a, b$ $U(-1, 1)$ and $c$ $U(-1/m, 1/m)$. Thus, as we scale down the numberof neurons, the value of $\delta$ grows, making the network function adapt more to the data. Figure 7 shows the results of this experiment.

Figure 7: The effect of varying the number of neurons $m$ the top image uses 20 neurons, the middle uses 200 and the bottom uses 2000. Observe that with fewer neurons, the function is adaptive to the data since $\delta$ gets larger.

### E.1 Visualizing Attractor Samples

We can visualize the vector field $(\partial_t(\hat{r}, \theta)$ by considering the change of metric from $w = (\hat{r}, \theta)$ to $(u, v)$ with the map

$$\pi_{(u,v)}(\hat{r}, \theta) = (|\hat{r}| \cos \theta, |\hat{r}| \sin \theta) = (u, v).$$

Assuming we know the sign of $\hat{r}$, the vector field

$$\begin{bmatrix} \partial_t u \\ \partial_t v \end{bmatrix} = D\pi_{(u,v)} D\pi_{(u,v)}^T \begin{bmatrix} \partial_t r \\ \partial_t \theta \end{bmatrix} \tag{36}$$

Observing that $D\pi_{(u,v)} D\pi_{(u,v)}^T = I$, we have simply that

$$\begin{bmatrix} \partial_t u \\ \partial_t v \end{bmatrix} = \begin{bmatrix} \partial_t r \\ \partial_t \theta \end{bmatrix}$$

Figure 8 shows a plot of this vector felt by a single particle in $uv$ in the case where $\delta = \infty$. In this case, the partial derivative $\partial_t r$ remains unchanged. Furthermore, we remark that at the boundaries of samples, the vector field can change directions, causing these samples to become "attractors" or "repulsors" (see Lemma 1 in the main document).

## F  Implicit Regularization of the Kernel Regime

We can write the network function (1) in the standard parameterization as a matrix product:

$$f_{\boldsymbol{z}}(x) = M\boldsymbol{c}, \qquad M \in \mathbb{R}^{s \times m}, \qquad M_i j = [a_j x_i + b_j]_+ \tag{37}$$

Now consider the least-squares problem:

$$\text{minimize } \frac{1}{2}||M\boldsymbol{c} - \boldsymbol{y}||^2 \tag{38}$$

If we are in the kernel regime (see Section 3.3) then we minimizing (38) by only changing the parameter $\boldsymbol{c}$. Following gradient flow, we have that $\boldsymbol{c}$ follows the ODE

$$\partial_t \boldsymbol{c}(t) = -\nabla \frac{1}{2}||M\boldsymbol{c}(t) - \boldsymbol{y}||^2 \tag{39}$$
$$= -(M^T M\boldsymbol{c} - M^T \boldsymbol{y}). \tag{40}$$

In the finite width case, the kernel $K$ can be written as

$$K(x_i, x_j) = K = (MM^T)_{ij}$$

And, thus, $f(t) = M\boldsymbol{c}(t)$ follows the separable ODE:

$$\partial_t f(t) = M\partial_t \boldsymbol{c}(t) \tag{41}$$
$$= MM^T \boldsymbol{y} - MM^T M\boldsymbol{c} \tag{42}$$
$$= Ky - KM\boldsymbol{c} \tag{43}$$
$$= K\boldsymbol{y} - Kf(t) \tag{44}$$

whose solutions are of the form:

$$f(t) \propto \exp(-K)\mathbb{1}t + y \tag{45}$$

We can decompose $K$ into its eigenbasis where $\mathbf{e_i}$, an eigenvector of $K$ and $\lambda_i$ is the corresponding eigenvalue for $i = 1, \ldots, s$.

We can then write (45) as:

$$f(t) = y + \sum_{i=1}^n exp(-t\lambda_i \mathbf{e_i}) \tag{46}$$

Figure 8: *Top:* The gradient field ([18](navigation)) felt by a particle. Note how the vectors change directions at certain samples. These samples are "attractors" or "repulsors" where particles get stuck or get pushed away from. *Bottom Left:* A plot of the network function for the gradient field in the top image. Observe how there are clusters of neurons (blue circles) aligned with certain samples. *Bottom Right:* A plot of the neurons in $uv$ space. Observe how the red neurons cluster at "attractor" points in the top image

which implies the dynamics of the residual, $f(t) - y$ are:

$$f(t) - y = \sum_{i=1}^{n} exp(-t\lambda_i \mathbf{e_i}) \tag{47}$$

Thus, as $t \to \infty$, the terms $exp(-t\lambda_i \mathbf{e_i})$ decay, at a rate exponential in $\lambda_i$ and $f(t) \to y$, which, as mentioned in [16], suggests that early stopping acts as a regularizer by decaying the residual primarily along the principle components of the kernel. These principle components usually correspond to smoother functioons. Furthermore, (47) means that if the kernel is not full rank, then the residual cannot go to zero since some $\lambda_i$ will be zero.