[Reviews · NeurIPS 2019]

Reviewer 1



After rebuttal: I have carefully read the comments of other reviewers and the feedback from the authors. The authors provide insights about generalization to high dimensional cases promise to include the experimental results for high dimensions, which address my main concern. However, the experiment results are not provided in the rebuttal. Thus, I would like to keep my score unchanged. -------------------------------------------- Summary: This paper analyzed the dynamics of gradient descent algorithm training an over-parametrized one-hidden-layer ReLU neural network on an interpolation task in three settings and compare them. In this paper, the authors are using a neural network with one hidden layer and one output layer. The activation function is ReLU, and each hidden unit also has a bias term that can be trained. The input is 1-dimensional, i.e., a scalar, so the task is equivalent to interpolation. The loss function is square loss, and the authors use gradient descent to learn the network weights. The authors use an over-parametrization scheme where the number of nodes in the hidden layer tends to infinity, which means the network is infinitely wide. There are two main learning schemes mentioned in this paper: Kernel and adaptive. The Kernel scheme is that the weights of the neural network stay very close to the initialization and the optimization process works in a linear approximation in this neighborhood. The adaptive scheme is using mean-field setting and the evolving of the weights follows a PDE. In the adaptive scheme, since ReLU function is positively homogeneous, the weights have some redundant terms. Thus, the authors divide the learning scheme into the reduced parameters version and full version. The authors identified a condition \delta, which only depends on initialization, that decides the style of the learning process. If \delta is very large, we are in adaptive scheme, resulting in adaptive linear splines, while when \delta is very small we are in the Kernel scheme, resulting in cubic splines. Otherwise, we are somewhere in the middle of these two schemes. The authors also characterize the dynamics in these two schemes, and their experimental results validate their theoretical analysis. My comments for this paper: 1. The experiments in this paper focused on the exact setting as the theoretical analysis, but the theoretical analysis doesn't provide much intuition about whether these kinds of phenomena can generalize to more complicated cases like high-dimension case. Thus, it would be better if the authors could do some experiments about general cases and whether some similar dynamics preserves. 2. For the Kernel methods, some recent results showed that even if the hidden layer is only polynomially wide, the weights will still stay in a small neighborhood with high probability. I wonder whether in this case the dynamics are the same as the result in this paper.

Reviewer 2



The authors study the inductive bias of relu networks in dimension 1. They distinguish between the active and kernel regime, which is a very timely question in deep learning. This paper gives a very satisfactory and precise answer for the kernel regime. Theorem 4 shows that in 1 dimension the kernel method interpolates according to a cubic spline. Unfortunately , the result for the adaptive regime are not clear. Theorem 3 simply states some dynamics and then remarks below that if c^2>a^2+b^2 then the dynamics follow the reduced dynamics. However I do not see that they formally proved it follows an adaptive linear splines. I have read the responses and the authors seem to address some of my comments. I'll keep my score at "marginally above"

Reviewer 3



Although the result in this paper only applies to one-hidden-layer networks with one dimension input, it is an extremely important direction of understanding the difference between neural network learning and kernel learning. Especially under the current hype about neural tangent kernel (NTK). The main concern for me about this work is the paper is written in a non-rigorous way, which makes it hard to understand the technical contribution of the work. For example, the main claim "in adaptive learning regime, the learning favors linear splines" (as in abstract) is never formally proved anywhere. The only Lemma in this adaptive regime is Lemma 1, however, the Lemma is more an "intuitive statement" than a formal theorem: the \rho in the Lemma is the \rho at some iteration, and it is not clear what is the picture about the full dynamic when all the iterations are taking into consideration. (for example, a point can be attractor in one iteration and repulsor for another). It is possible that in the long ran all these attractor/repulsors get canceled and the network does not adapt to the data eventually. The smaller concern is that the paper is missing some proper citations about previous works in learning under over-parameterization, mostly the kernel regime: Learning Overparameterized Neural Networks via Stochastic Gradient Descent on Structured Data A Convergence Theory for Deep Learning via Over-Parameterization Learning and Generalization in Overparameterized Neural Networks, Going Beyond Two Layers Especially the second work actually proves that the neural network learns the solution of the best kernel in the "kernel learning regime", even for deep networks. The authors cited some follow-up papers of these works but the original ones are not properly addressed. After Rebuttal: The authors have promised to improve their paper in the next revision. Their promises make sense and addressed my concerns. I am willing to higher my score.

[Author Response · NeurIPS 2019]

We thank the reviewers for their insightful comments and suggestions. We respond to the major concerns below and
will incorporate all comments in the next revision.

**Summary of contributions** We presented a novel theoretical and empirical study of the gradient dynamics of overpa-
rameterized shallow ReLU networks trained with a least-squares loss. Our results are valid both in the finite and infinite
width functional settings. We distinguish two extremal regimes in terms of generalization behavior: "adaptive" and
"kernel". The effect of each regime can be quantified in terms of a conserved quantity which depends on the initialization
and on the scaling as the number of neurons grows large. In the kernel regime, the training problem converges to a
kernel regression over a Sobolev space $\mathcal{H}^{2,2}$ as the number of neurons approaches infinity. Furthermore, in 1D, under
mild technical assumptions, the kernel case reduces to cubic spline interpolation. In the mean-field limit, the adaptive
regime with regularization converges to regression in $\mathcal{H}^{1,2}$, yielding linear splines with knots at the samples. For a finite
number of neurons, our presentation of the adaptive regime is qualitative. We note that dynamics are fully determined
by the residuals and the velocity field induced by gradient flow always pushes neurons towards the samples. For finite
neurons, we observe solutions which adapt to the input data with knots converging at samples.

**Analysis of the adaptive regime (R2, R3)** Our results on the adaptive regime reinterpret those appearing in *[Maennel*
*et al.]* and *[Savarese et al.]* in the framework of mean-field analysis. The functional representation in terms of linear
splines is established in the limit of infinite width (by combining *[Saverese et al.]* with *[Chizat and Bach NeurIPS'18]*)
under appropriate initial conditions and using TV regularisation. Our analysis in the adaptive regime for finite neurons
is thus qualititative, but we believe it clarifies the role of initialization and parametrization. Rigorously quantifying the
effect of having a finite number of neurons is an important next step, as is the extension to other neural architectures.
We highlight however that our main technical contribution in this work is to rigorously establish the implicit bias in
the kernel regime in terms of cubic splines, for generic parameter initializations. We therefore provide one of the first
instances of an explicit distinction between the "adaptive" and the "kernel" regimes in terms of generalization: formally,
we can show that kernel training converges to a kernel regression in $\mathcal{H}^{2,2}$ and, following *[Chizat and Bach NeurIPS'18]*,
that adaptive training in the mean-field limit converges to a regularised regression in a Sobolev space $\mathcal{H}^{1,2}$. If the paper
is accepted, we will emphasize these technical contributions.

**Insights into higher dimensional inputs (R1)** The statements and formulation in the paper generalize to higher
dimensional inputs, however they do not paint a complete picture of the dynamics in this setting. For higher dimensional
full parameters $(\boldsymbol{a} \in \mathbb{R}^{m \times p}, \boldsymbol{b} \in \mathbb{R}^m, \boldsymbol{c} \in \mathbb{R}^m)$ representing $f_{\boldsymbol{z}} : \mathbb{R}^p \to \mathbb{R}$, the associated reduced parameters can be
viewed as spherical coordinates identifying each neuron with a unit-norm vector $||d(\boldsymbol{\theta}_i)|| = 1$ in $\mathbb{R}^{p+1}$ and a radius
$r_i = c_i ||(\boldsymbol{a}_i, b_i)||_2$.

In higher dimensions, the samples correspond to hyperplanes in phase space, and the possible configurations of attractors
and repulsors become more complex. For example, when reduced neurons lie on one of the attractor hyperplanes,
they follow dynamics in the lower dimensional subspace. The difficulty with the analysis in higher dimensions is
that it involves the combinatorics of arrangements of hyperplanes corresponding to the sample points. We leave full
categorization of these dynamics to future work, however we were able to verify experimentally that the dynamics in
higher dimensions are qualitatively very similar to the 1D case, leading to concentration of neurons in the adaptive
regime and smooth interpolants in the kernel regime. If the paper is accepted, we will include these experimental results.

**Definition of linear splines (R2, R3)** We will give a formal definition of adaptive linear splines in the next revision of
the paper: A linear spline is a piecewise linear function $\varphi : \mathbb{R} \to \mathbb{R}$ whose knots $e_i \in \mathbb{R}, i = 1 \dots m$ are the boundaries
between pieces. We say that the spline is adaptive if the knots are also variable, i.e., if the function can be written
as $\varphi(x', e_1, \dots, e_m) : \mathbb{R} \times \mathbb{R}^m \to \mathbb{R}$. Alternatively, we can view adaptive linear splines in the funtional setting as
functions $\varphi : \mathbb{R} \to \mathbb{R}$ which interpolate the data points and minimize $\|\varphi\|_{\mathcal{H}^{1,2}} := \int |\varphi''(u)| du$.

**Kernel learning for polynomially wide networks (R1)** We were not aware of these results for polynomially wide
networks, and will cite them in revised version of the paper. These, in fact, seem complementary to our results which
demonstrate that for increasing width, the dynamics rapidly approach the kernel regime. In the revision, we will include
an experiment demonstrating results for varying finite widths.

**Missing citations (R3)** The missing citations pointed out by the reviewer are relevant and will be addressed in the
next revision. In particular we believe our work is complementary to "A Convergence Theory for Deep Learning via
Over-Parameterization" since we can quantify for both a finite and infinite number of neurons how much the dynamics
behave like the kernel regime versus the adaptive regime by considering $\boldsymbol{\delta}$ and $m$ in Equation (21).

**Presentation of the results (R3)** We have prepared a revised version of the paper where our results are presented
more rigorously. In particular, we have improved Proposition 4 and formalized our discussion relating the RKHS
norm with linearized curvature. In the adaptive setting, in addition to the infinte width analysis, we have clarified the
qualitative description of the dynamics with finite neurons and added experiments illustrating the role of attractive
samples throughout the training process (in 1D and in higher dimensions).

[Meta-Review · NeurIPS 2019]

The paper studies the training dynamics of 1D ReLU networks with one hidden layer in the kernel regime and the adaptive regime, and shows connections with spline interpolation. The reviewers all appreciated the importance of the contribution in the kernel regime, but found that the analysis of the adaptive regime is not as clear/rigorous. Still, there is an overall consensus that the paper is above the bar for acceptance.